# Simulation of Soil Cutting and Power Consumption Optimization of a Typical Rotary Tillage Soil Blade

**Xiongye Zhang, Lixin Zhang \*, Xue Hu, Huan Wang, Xuebin Shi and Xiao Ma**

School of Mechanical and Electrical Engineering, Shihezi University, Shihezi 832003, China
* Correspondence: zhlx2001329@163.com

**Abstract:** The rotary tillage knife roller, as one of the typical soil-touching parts of the tillage equipment cutting process, is in direct contact with the soil. During the cutting process, there are problems related to structural bending, deformation, and high power consumption, caused by impact and load, and it is difficult to observe the micro-change law of the rotary tillage tool and soil. In view of the above problems, we took the soil of the cotton experimental field in Shihezi, Xinjiang, and the soil-contacting parts of the rotary tillage equipment, specifically the rotary tiller roller, as the research subject. Using the finite-element method (FEM) to simulate the structure of the rotary tiller with different bending angle parameters, we obtained its average stress and deformation position information, and obtained a range linear relationship between the bending angle and the structural performance of the rotary tiller tool. Using discrete element method (DEM)-based simulation to build the corresponding contact model, soil particle model, and soil–rotary tillage knife roll interaction model to simulate the dynamic process of a rotary tillage knife roll cutting soil, we obtained the change rules of the soil deformation area, cutting process energy, cutting resistance, and soil particle movement. By using the orthogonal simulation test and the response surface method, we optimized the kinematic parameters of the rotary tiller roller and the key design parameters of a single rotary tiller. Taking the reduction of cutting power consumption as the optimization goal and considering the influence of the bending angle on its structural performance, the optimal parameter combination was obtained as follows: the forward speed was 900 m/h, the rotation speed was 100 rad/min, the bending angle was 115°, and the minimum power consumption of the cutter roller was 0.181 kW. The corresponding average stress and deformation were 0.983 mm and 41.826 MPa, which were 15.8%, 13%, and 7.9% lower than the simulation results of power consumption, stress, and deformation under the initial parameter setting, respectively. Finally, the effectiveness of the simulation optimization model in reducing power consumption and the accuracy of the soil-cutting simulation were verified by a rotary tilling inter-field test, which provided theoretical reference and technical support for the design and optimization of other typical soil-touching parts of tillage and related equipment, such as disc harrow, ploughshare, and sub-soiling shovel.

**Keywords:** rotary blade roller; FEM; DEM; simulation; response surface method; optimize the power consumption

---

## 1. Introduction

Tillage and soil preparation is an important part of agricultural production, and the rotary cultivator is an important piece of tillage equipment. Its component, the rotary tillage knife roller, directly contacts the soil and is a key working part to achieve efficient rotary tillage operation [1]. The simulation of the rotary tillage process is of great significance for exploring the interaction between agricultural tillage machinery and field soil, and for optimizing the design and development of soil-contacting agricultural machinery. With the development of computer-aided engineering in the late 20th century, researchers gradually applied numerical simulation and virtual simulation methods to agricultural machinery interacting with soil [2]. Especially in the study of typical soil-touching parts such as

the ploughshare, rotary tiller, disc harrow, and so on [3–5]. This simulation method can produce fast prediction without many field tests, but its simulation model and equation still need to be improved to improve its accuracy. The discrete-element method (DEM) has enabled research progress in simulating the deformation between granular materials and research materials, such as material parameter calibration [6], crop cultivation simulation, and so on [7,8], in agricultural engineering. The finite-element method (FEM) has high accuracy and efficiency in the calculation of mechanical deformation of continuum medium with small size structure [9]. Domestic and foreign scholars such as Yeon soo Kim [10] used discrete element software to model agricultural soil and predict the traction force at different tillage depths. The DEM soil model was calibrated by virtual blade shear test, and the field traction force was tested. The prediction error was less than 7.5%. Sun Jingbin et al. [11] studied the law of soil disturbance on slope by rotary cultivator using the discrete-element method and determined the primary and secondary factors affecting soil lateral displacement through variance analysis. Xiongpingyuan et al. [11] and Fang Huimin et al. [12] established the discrete-element simulation model of rotary tiller soil in the study area. Xiong Pingyuan et al. [11] and Fang Huimin et al. [12] established the rotary tiller soil discrete-element simulation model in the study area to analyze the three-dimensional working resistance of the rotary tiller when working in the soil with or without straw coverage. The relative error between the test value and the simulation value is within the reasonable range of 11–19%. Liu Qianwen et al. [13] established the strength analysis model of the rotary tillage knife using the finite-element method, and used the element grouping method to analyze the dynamic stress and power of soil cutting. Yang Hongkun et al. [14] carried out structural and modal simulation on stress concentration area and deformation of a fertilizing trenching plough through Ansys. They found that it can improve the strength and reduce the structural deformation of the plow shovel and plow post under constrained mode, and meet the requirements of working intensity power consumption optimization in the agricultural machinery soil tillage process. For example, Zhang Pilohang [15] demonstrated drag reduction and consumption reduction in the rotary tillage process by optimizing the arrangement and combination of rotary tillage knife roller blades. Li Shoutai [16] used an orthogonal rotation combination experiment and response surface optimization method to obtain the optimal combination of key structural parameters, and verified the effectiveness of power consumption optimization through a soil tank test. Han Yujie [17] determined the optimal combination of working parameters through the optimization of working parameters, such as speed and tillage depth, to achieve reduced cutting power, and verified these parameters by field experiments. In the simulation and optimization study of the above soil-touching components, the simulation of the dynamics of the rotating tiller roller is not fully considered. Most studies do not use structural performance as an evaluation index and finite-element simulation as verification. The simulation model and established simulation accuracy need to be improved. Additionally, in the optimization study, the type of optimal factors is relatively singular, usually without considering its own structural parameters.

This paper takes the rotary blade roller as the specific research subject and combines the advantages of the two simulation methods to carry out numerical simulation of the rotary blade. The structure of the rotary tiller was simulated and analyzed using ANSYS software to obtain its structural performance information, such as stress and deformation. The discrete-element EDEM software was used to simulate the soil tillage process by rotary blade roller, and the variation laws of cutting resistance and soil micro-motion trajectory were obtained and analyzed. The combined experimental optimum design method, based on the finite-element structural analysis and discrete element soil cutting dynamics analysis, the key structural parameters of the rotary blade, and the working motion parameters of the rotary blade roller were optimized to obtain the optimal parameter combination and reduce the actual cutting power consumption.

## 2. Principle of Operation and Materials

### 2.1. Model Structure and Motion Analysis

In order to optimize the structural parameters of the rotary blade, parametric modeling was carried out in SOLDWORKS. In this simulation test, the rotary blade of the 1GQK-125 rotary component was used as the reference for modeling, and the schematic diagram of its rotary blade is shown in Figure 1.

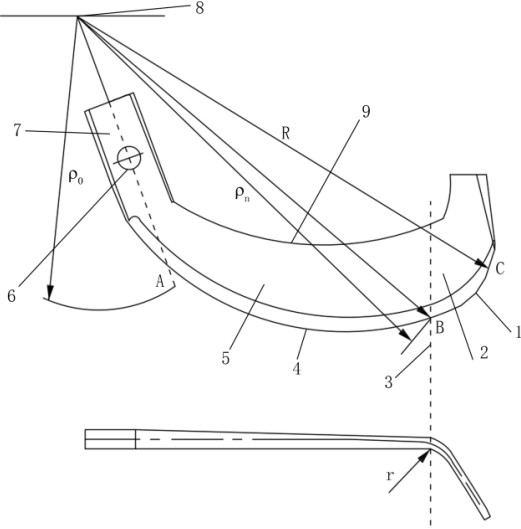

**Figure 1.** Schematic diagram of rotary blade structure. 1. positive cutting edge; 2. front cutting part; 3. bending line; 4. side cutting edge; 5. side cutting part; 6. mounting hole; 7. tool handle; 8. rotary center of knife roller; 9. curve of back edge.

In order to realize dynamic simulation of the rotary blade in a discrete element simulation, a three-dimensional model of the rotary blade was established. Its three-dimensional assembly is shown in Figure 2, which consists of left (right) rotary blade, knife holder, and roller shaft parts.

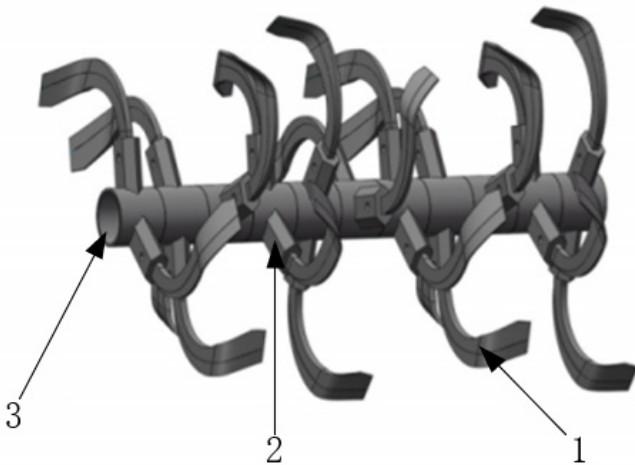

**Figure 2.** Structure of rotary tiller roller 3D model. 1. roller; 2. knife seat; 3. rotary tillage single knife.

The analysis of the rotary tiller motion process provides a reference for setting boundary conditions for subsequent dynamic simulations of soil cutting. The rotary cutter soil-cutting movement process mainly includes two parts: (1) with the rotary tiller forward direction of motion, that is, the rotary blade forward horizontal speed; (2) the rotational movement around the rotary cutter axis (after the pendulum line), that is, the angular

velocity of the rotary cutter, according to the definition of the soil cutting motion coordinate system, as shown in Figure 3 [18]. The trajectory of the rotary tillage elbow can be expressed as:

$$\begin{cases} x = R\cos(\omega t) + V_m t \\ y = R\sin(\omega t) \end{cases} \tag{1}$$

where $\omega$ is angular velocity of rotary blade rotation (rad/s); $t$ is soil cutting time (s); $V_m$ is the speed of rotary blade at a certain time (m/s).

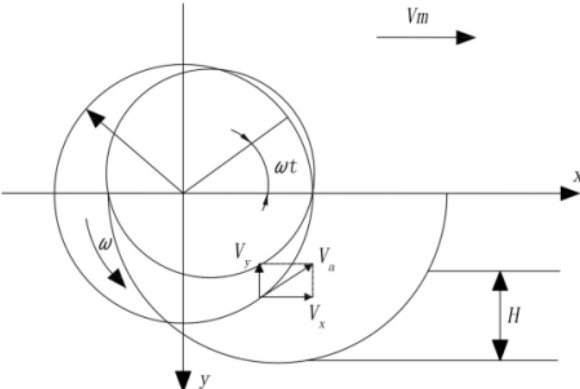

**Figure 3.** Rotary cutter end-point trajectory.

*2.2. Determination of Simulation Material Parameters*

The simulation parameters are divided into material intrinsic parameters and contact parameters [19–21], and the material intrinsic parameters are the characteristic parameters of the material itself. The materials used in this paper are public and widely used materials, which are obtained by referring to national standards and literature [22,23]. The material contact parameters can be obtained by experimental measurement or simulation test calibration. In this paper, the soil accumulation angle test and sliding test were used. Since Shihezi in Xinjiang has sandy soil, the JKR surface energy is not considered. The specific parameters of simulation are shown in Tables 1 and 2.

**Table 1.** Material intrinsic parameter.

| Material | Density/(kg·m$^{-3}$) | Poisson's Ratio | Shear Modulus/Pa |
|---|---|---|---|
| 65Mn-stell | 7850 | 0.37 | $0.25 \times 10^{10}$ |
| Soil | 1250 | 0.36 | $1.0 \times 10^{6}$ |

**Table 2.** Simulated contact parameters.

| Parameter | Soil–Soil | Soil–Tool/65Mn |
|---|---|---|
| Recovery coefficient | 0.43 | 0.52 |
| Coefficient of rolling friction | 0.50 | 0.06 |
| Coefficient of static friction | 0.24 | 0.3 |

## 3. Finite-Element Structural Simulation

*3.1. Finite-Element Simulation Method*

As one of the key structural design parameters of the rotary tiller [24], the selection of its parameters has an important impact on its overall cutting performance. Three parameters were selected for modeling: 110°, 120°, and 130°. As shown in Figure 4, the influence of the bending angle on its performance was analyzed.

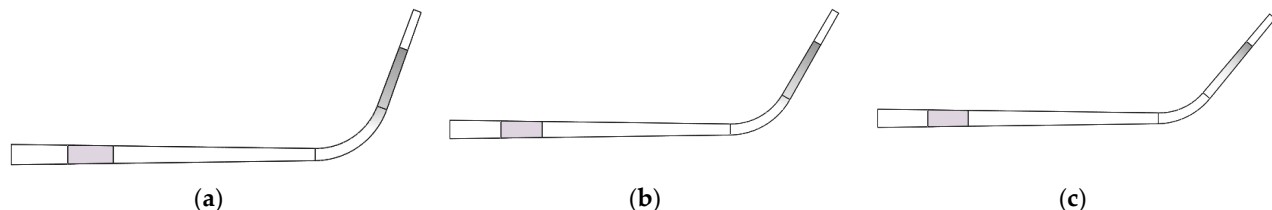

(**a**)　　　　　　　　　(**b**)　　　　　　　　　(**c**)

**Figure 4.** Rotary tiller with different bending-angle models. (**a**) bending angle of 110°; (**b**) bending angle of 120°; (**c**) bending angle of 130°.

### 3.2. Simulation Results of Finite-Element Structure

ANSYS finite-element software was used to simulate the structural performance of rotary blade parts with bending angles of 110°, 120°, and 130° to carry out pre-processing, such as grid division and boundary condition definition. Tetrahedral mesh was used to divide the geometry automatically; the size of the mesh was set to 6 mm. Load and boundary conditions were added to the rotary cutter according to the actual cutting conditions. Fixed constraint was added to the inner surface of the circular mounting hole, and a force of 500 N was applied perpendicular to the surface direction at the side cutting edge, at the positive cutting edge, and at the edge position of the transition surface of the rotating curved knife.

The deformation region and numerical cloud diagram were calculated, as shown in Figure 5. The average deformations calculated by simulation were 0.94 mm, 1.13 mm, and 1.22 mm. The average stress was 39.67 Mpa, 45.39 Mpa, and 50.98 Mpa. It can be seen that the stress and deformation increase with the increase of bending angle in the range of 110–130°, and the relationship can provide reference for subsequent structural parameter optimization.

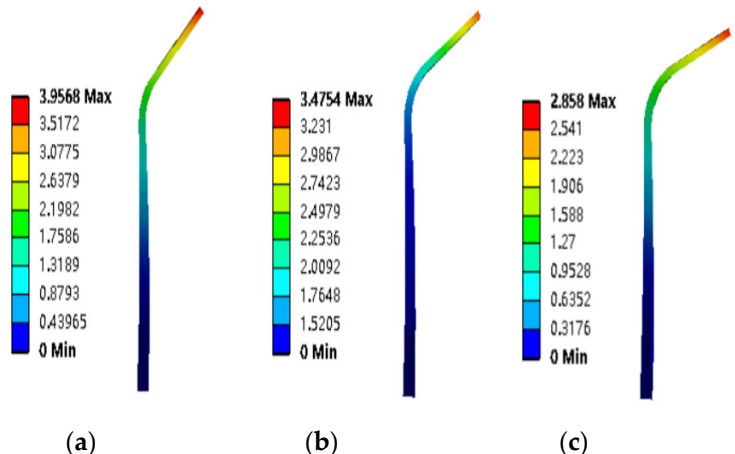

(**a**)　　　　　　(**b**)　　　　　　(**c**)

**Figure 5.** Stress clouds for different bending angles of rotary cutter. (**a**) 130° (**b**) 120° (**c**) 110°.

## 4. Discrete-Element Dynamic Simulation

### 4.1. Contact Model and Soil Template Establishment

By measuring the field soil solid rate, moisture content, and other data via access to the China Soil Database [25] it was determined that the test soil was gray desert loess, in soil particle size and shape. Plastic material was chosen, according to the test soil types, to create a delayed elastic contact model (hysteretic spring contact model, HSCM). The contact model between the particles of the physical model is shown in Figure 6a. The model allows the plastic deformation behavior to be added to the contact mechanics equation, so that, since a predefined stress particle displays an elastic behavior, once the stress is exceeded, the particle behaves as if it is undergoing plastic deformation. The result is a large overlap without excessive force to characterize a compressible material. The exact location of each

old contact point is not "remembered", so the particles become undeformed once they are separated. Subsequent contacts will be loaded along the new loading route of slope $K_1$, and any reloading prior to separation is loaded at slope $K_2$ until the original loading curve $K_1$ is reached, followed by this slope until unloading occurs, the schematic diagram of the model force–displacement relationship is shown in Figure 6b.

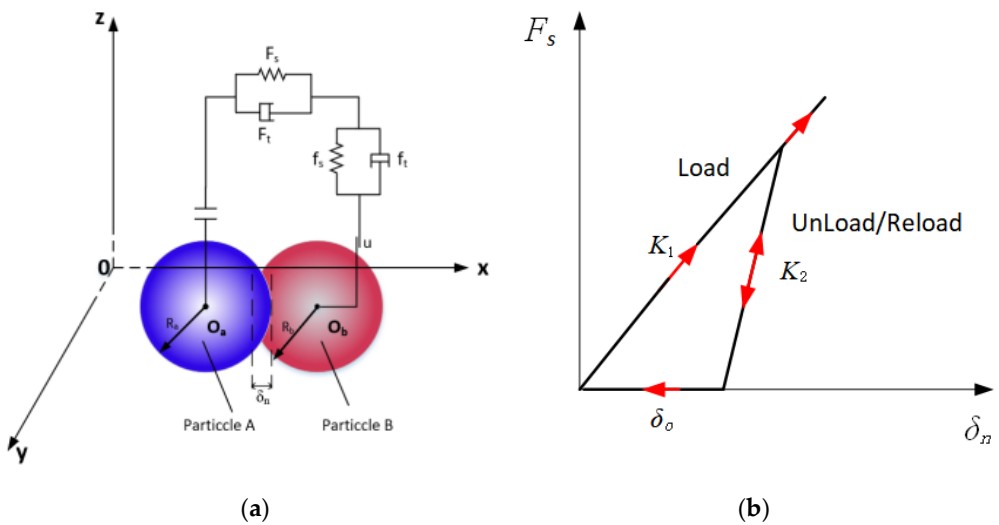

| (a) | (b) |

**Figure 6.** HSCM Contact Model. (**a**) Particle–contact relationship; (**b**) Force–displacement relationship. Note: $O_a$ and $O_b$ are the spherical center positions of two particles; $R_a$ and $R_b$ are the radii of two particles, mm; $\delta_n$ is the normal overlap of particle collision, mm; $\delta_0$ is the residual overlap between particles, mm; $F_s$ and $F_t$ are the normal contact force and damping force, N; $f_s$ and $f_t$ are the tangential contact force and damping force, N; $\mu$ is the friction coefficient.

The HSCM normal force $F_N$ is calculated using the following equation:

$$F_N = - \begin{cases} K_1\delta_n & (K_1\delta_n < K_2(\delta_n - \delta_0) \\ K_2(\delta_n - \delta_0) & (\delta_n > \delta_0) \\ 0 & (\delta_n \leq \delta_0) \end{cases} \tag{2}$$

where $K_1$ and $K_2$ are the loading and unloading stiffnesses, respectively; $\delta_n$ is the normal overlap; and $\delta_0$ is the residual overlap. The loading stiffness, $K_1$, is related to the yield strength of each material involved in contact. The relationship between $Y_1$ and $Y_2$ is as follows:

$$K_1 = 5R^*\min(Y_1, Y_2) \tag{3}$$

where $R^*$ is the equivalent radius of two contact particles, and $Y_1$ and $Y_2$ are the yield strengths of particles a and b, respectively.

The following expression for the recovery factor can be used for $K_2$ to determine $K_1$:

$$e = \sqrt{\frac{K_1}{K_2}} \tag{4}$$

where $e$ is the inter-particle recovery coefficient.

The following expression for the recovery factor can be used for $K_2$ to determine $K_1$ The amount of residual overlap is updated at each time step according to the following law:

$$\delta_0 = \begin{cases} \delta_n(1 - K_1/K_2) & (K_1\delta_n < K_2(\delta_n - \delta_0) \\ \delta_0 & (\delta_n > \delta_0) \\ \delta_n & (\delta_n \leq \delta_0) \end{cases} \tag{5}$$

The main energy-dissipation mechanism is the difference in spring stiffness between the loaded and unloaded phases.

For the establishment of soil particles based on a simplified spherical model by EDEM, a total of three kinds of soil particles in the shape of the model was established. As shown in Figure 7 (a) single-sphere model with a radius of 6 mm; (b) double-sphere model, with a diameter of 8 mm after combination; (c) linear three-sphere model with a diameter of 9 mm after combination.

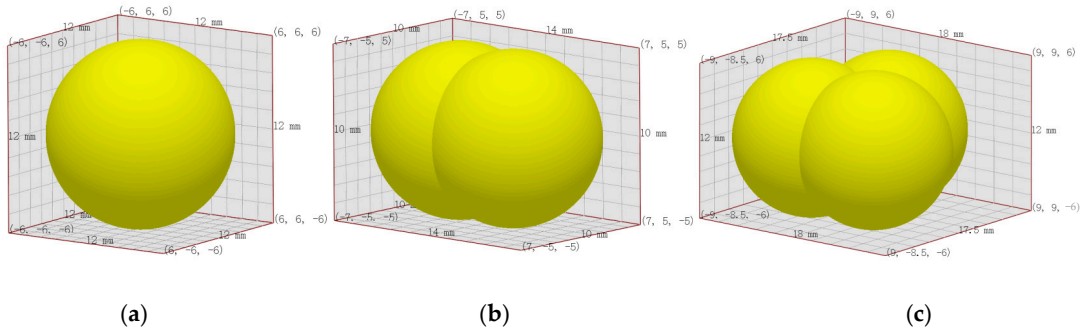

(**a**)　　　　　　　　　　　　　　　　(**b**)　　　　　　　　　　　　　　　　(**c**)

**Figure 7.** Soil particle template. (**a**) radius of 6 mm; (**b**) radius of 8 mm; (**c**) radius of 9 mm.

### 4.2. Discrete-Element Model of Soil-Rotary Blade Roller

According to the cutting mode and boundary conditions of the rotary tillage cutter roll, a 1200 × 600 × 250 mm cuboid soil trough model was constructed as follows: a 1200 × 600 × 250 mm cuboid soil trough model was constructed and a virtual surface was built on top of it, then $1.8 \times 10^6$ soil particles were generated to fill the soil bin to realize the soil environment simulation. At the same time, the cutter roll was set to rotate the soil counterclockwise, and the forward speed $v$ = 1100 m/h, and the rotational speed $n$ = 20 r/min were defined as consistent with the test. The cutter roll–soil groove interactive discrete- element model is shown in Figure 8.

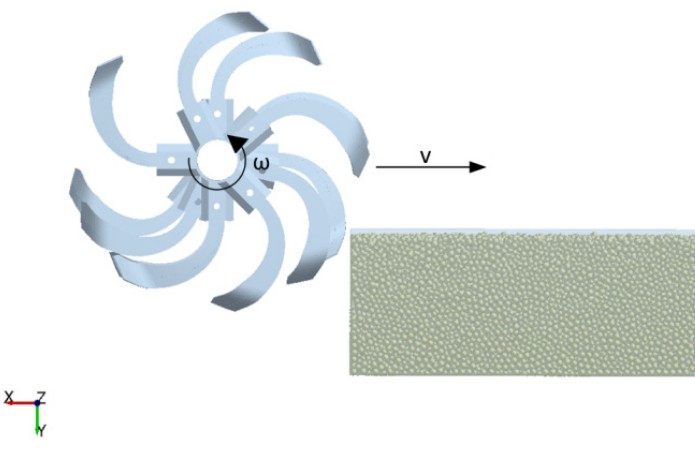

**Figure 8.** Discrete-element interaction simulation model.

### 4.3. Simulation Results of Discrete Element Dynamics

4.3.1. Simulation Results of Cutting Process

Through EDEM post-processing setup selection, we set the color of the movement speed of the soil particles under the tool. The velocity of soil particles at the position in contact with the tool was high, and the cutting soil tended to move upward along the rotary tillage knife. The cutting simulation process is shown in Figure 9.

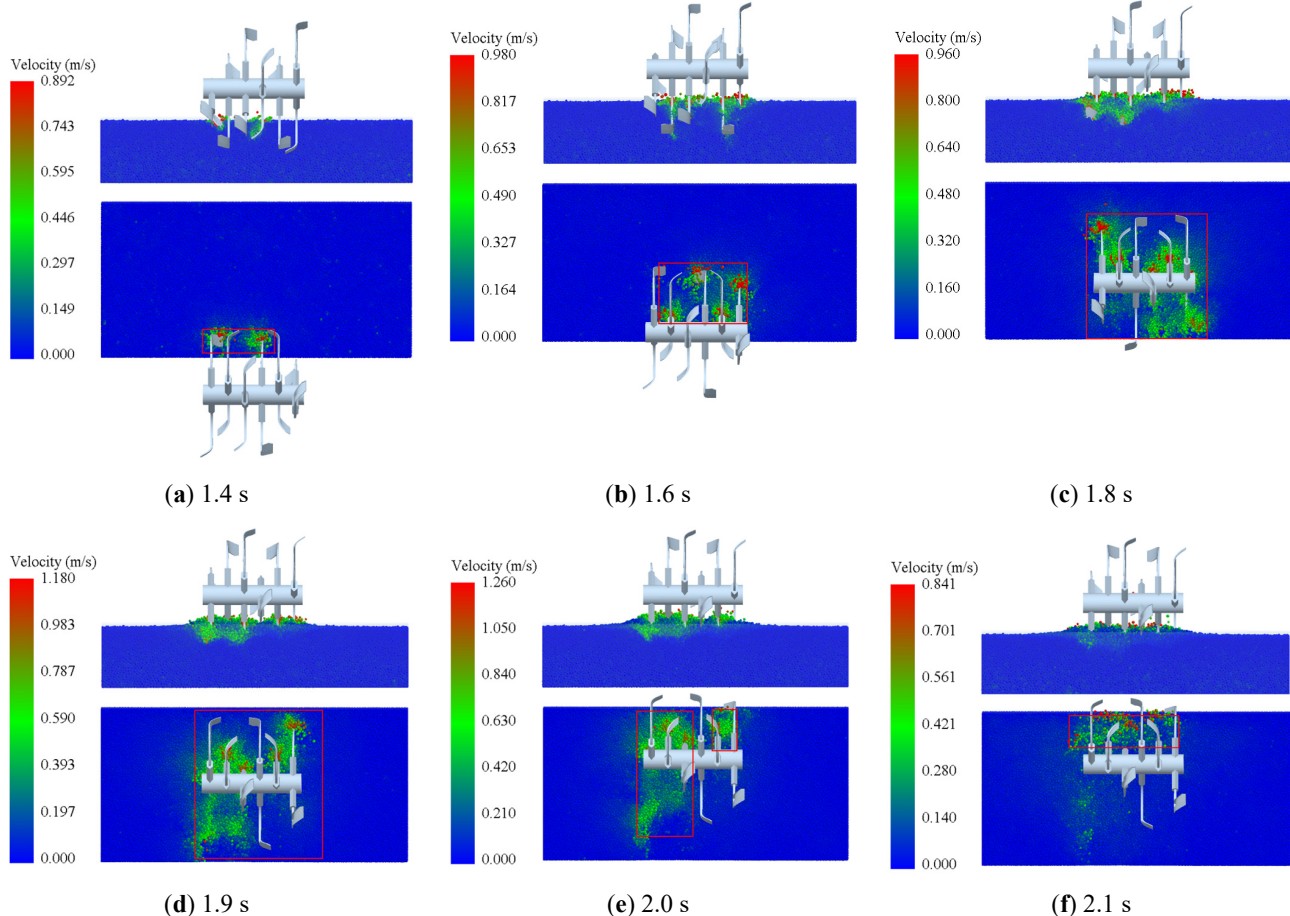

**Figure 9.** Cutting process simulation diagram.

In the process of the rotary blade roller gradually entering the soil, the tangent cutting edge of the rotary blade first contacted the soil. Then the soil was broken along the forward direction of the knife roll because the soil was pressed by the upper cutting edge of the rotary blade. The disturbed area of soil gradually increased (Figure 9a–c). Then, the soil was further damaged by the double action of the lateral cutting edge and the edges of multiple rotary tiller blades, and the phenomenon of scattering along the counterclockwise rotation direction of the knife roll appeared. Not until the knife roller entered the soil completely, did the soil disturbance area reach the maximum (Figure 9d). At the same time, this proved that the knife roller can push the soil longitudinally during the actual rotary tillage. Finally, as the knife roller gradually left, the soil disturbance area gradually decreased (Figure 9e,f).

### 4.3.2. Cutting Force Results and Analysis

The working resistance of the rotary blade roller is shown in Figure 10, and the variation rule of the torque of the blade shaft is similar to its trend. In the process of rotary tillage cutter roll cutting, when the tool did not contact the soil, the cutting resistance was 0. With the rotation of the roller, the rotary tillage cutter gradually contacted deep into the soil and the cutting resistance increased gradually. With the advancing and rotating motion of the rotary blade roller, the area and volume of the soil contacted by the blade increased, and the depth of the plow gradually increased. When the maximum depth of the plow was reached, the resistance of the blade roller also reached the maximum. When the maximum tillage depth was exceeded, the contact area of soil, the volume of soil cutting, and the corresponding resistance of the cutter roller decreased. The cutter roller rotates 720° within 1 s, so the cutting resistance of the rotary cutter roller presents two periodic

changes. These results are consistent with the theoretical analysis under the actual setting parameters, further validating the simulation results. Subsequent rototill field tests will be conducted for soil disturbance observation and comparison to verify the accuracy of the discrete-element simulation.

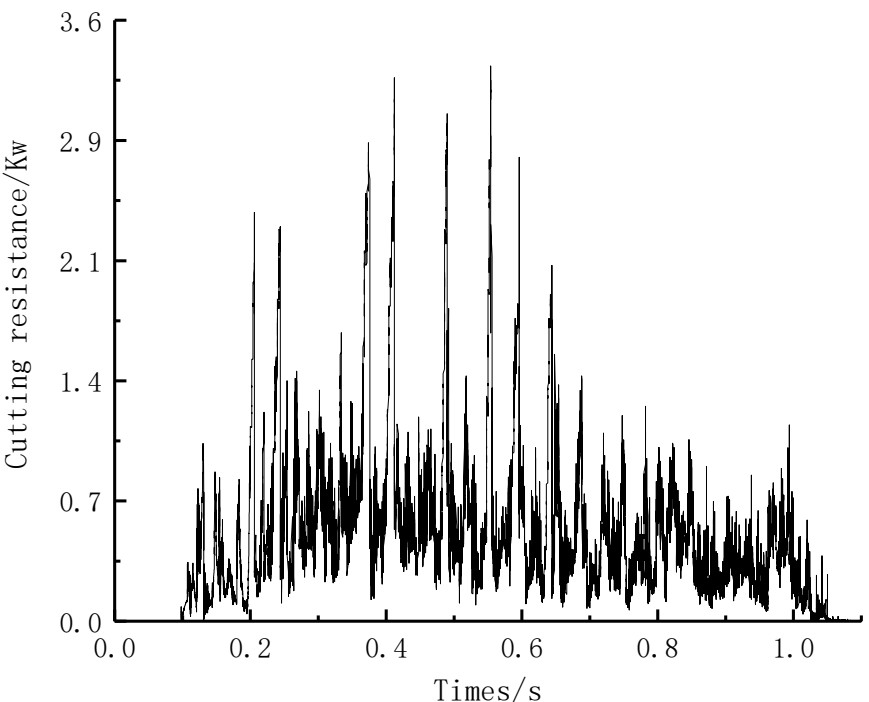

**Figure 10.** Cutting resistance curve.

## 5. Power Optimization

### 5.1. Power Consumption Optimization Test Method

The power consumption of a rotary tillage operation is influenced by many factors, such as motion parameters, the structure of the rotary tillage knife, cutting angle, cutting depth, soil properties, and so on. In order to achieve controllable simulation variables, this paper mainly explores the influence that roll motion parameters and rotary blade structure parameters have on the power consumption of two types of tools. In order to optimize the power consumption of the rotary blade/blade roller, there are three specific factors that influence the parameters. The forward speed ($X_1$) and rotation speed ($X_2$) are selected as the factors for the motion parameters, the bending angle is selected as the key structural design parameter affecting the cutting performance of the rotary blade [24], and the bending angle ($X_3$) is selected as the structural parameter optimization factor.

In this paper, the three-factor and three-level orthogonal test was used to conduct a simulation test with the cutting power consumption optimization index. The coding level of the test factors is shown in Table 3, and the simulation test results are shown in Table 4.

**Table 3.** Simulation test factor levels.

| | Factors | | |
|---|---|---|---|
| **Coding** | **Motion Parameters** | | **Structural Parameters** |
| | $X_1$<br>**Forward Speed m/h** | $X_2$<br>**Rotation Speed r/min** | $X_3$<br>**Bending Angle/°** |
| 1 | 1400 | 140 | 110 |
| 0 | 1100 | 120 | 120 |
| −1 | 800 | 100 | 130 |

**Table 4.** Design and results of cutting simulation experiment.

| Serial Number | $X_1$ | $X_2$ | $X_3$ | Cutting Power Consumption Y |
|:---:|:---:|:---:|:---:|:---:|
| 1 | −1.000 | −1.000 | 0.000 | 1.640 |
| 2 | 1.000 | −1.000 | 0.000 | 2.212 |
| 3 | −1.000 | 1.000 | 0.000 | 2.391 |
| 4 | 1.000 | 1.000 | 0.000 | 3.286 |
| 5 | −1.000 | 0.000 | −1.000 | 2.347 |
| 6 | 1.000 | 0.000 | −1.000 | 3.970 |
| 7 | −1.000 | 0.000 | 1.000 | 3.805 |
| 8 | 1.000 | 0.000 | 1.000 | 4.736 |
| 9 | 0.000 | −1.000 | −1.000 | 2.270 |
| 10 | 0.000 | 1.000 | −1.000 | 3.111 |
| 11 | 0.000 | −1.000 | 1.000 | 3.420 |
| 12 | 0.000 | 1.000 | 1.000 | 2.630 |
| 13 | 0.000 | 0.000 | 0.000 | 2.500 |
| 14 | 0.000 | 0.000 | 0.000 | 2.955 |
| 15 | 0.000 | 0.000 | 0.000 | 3.410 |
| 16 | 0.000 | 0.000 | 0.000 | 2.854 |
| 17 | 0.000 | 0.000 | 0.000 | 3.101 |

*5.2. The Optimization Results*

The F-test is a test under the null hypothesis (H0) where the statistical value follows the F-distribution, which is usually used to test the significance of the equation and the degree of influence of each factor on the test index. In this paper, Design-Expert software is used to process the test data in Table 4, test the fitting degree of the regression equation, verify the significance of each factor on the regression equation, and obtain the ANOVA results of cutting power consumption, as shown in Table 5. As can be seen from Table 5, model P (0.0052) is less than 0.01, indicating that the regression model is extremely significant. The misfitting term P (0.1961) was greater than 0.05, that is, the misfitting was not significant. And the goodness-of-fit of the regression model $R^2$ is 0.90, indicating that the quadratic regression equation fitted by the model is consistent with the reality, which can correctly reflect the relationship between cutting power consumption and $X_1$ (forward speed), $X_2$ (cutter roll speed), and $X_3$ (bending angle). The regression model can be used to predict the results of optimization tests. The regression equation of cutting power consumption was obtained after removing these insignificant factors:

$$Y = -2.96 + 0.5026X_1 + 0.3616X_3 - 0.4078X_2X_3 - 0.7193X_2^2 + 0.6130X_3^2 \tag{6}$$

**Table 5.** Analysis of variance of operating power.

| Source of Variation | Sum of Squares | Degree of Freedom | Mean Square | *f* Value | *p* Value | Significance |
|:---:|:---:|:---:|:---:|:---:|:---:|:---:|
| Model | 7.97 | 9 | 0.8851 | 4.81 | 0.0052 | ** |
| $X_1$ | 2.02 | 1 | 2.02 | 10.98 | 0.0290 | * |
| $X_2$ | 0.4399 | 1 | 0.4399 | 2.39 | 0.1660 | — |
| $X_3$ | 1.05 | 1 | 1.05 | 5.68 | 0.0486 | * |
| $X_1X_2$ | 0.0261 | 1 | 0.0261 | 0.1417 | 0.7178 | — |
| $X_1X_3$ | 0.1197 | 1 | 0.1197 | 0.6504 | 0.4465 | — |
| $X_2X_3$ | 0.6650 | 1 | 0.6650 | 3.61 | 0.0491 | * |
| $X_1^2$ | 0.0796 | 1 | 0.0796 | 0.4325 | 0.5318 | — |
| $X_2^2$ | 2.18 | 1 | 2.18 | 11.83 | 0.0108 | * |
| $X_3^2$ | 1.58 | 1 | 1.58 | 8.60 | 0.0220 | * |
| Residuals | 1.29 | 7 | 0.1841 | | | |
| Lack of Fit | 0.8433 | 3 | 0.2811 | 2.53 | 0.1961 | — |
| Pure Error | 0.4452 | 4 | 0.1113 | | | |
| Cor Total | 9.25 | 16 | | | | |

Note: $p \leq 0.01$ means extremely significant, marked with "**"; $p \leq 0.05$ means significant, marked with "*"; $0.01 < p < 0.05$ means insignificant, marked with "—".

According to the above analysis of variance and the regression coefficient of each factor in the regression model, each factor that has an influence on the cutting power consumption can be obtained. The obvious indigenity of each factor on the target optimization value is as follows: the first term $X_1$ and $X_3$, the interaction term $X_2X_3$, and the second term $X_2^2$ and $X_3^2$. The influence order is forward speed $X_1$ > bending angle $X_3$ > rotating speed $X_2$. The response surface based on the power consumption regression equation is shown in Figure 11. When the forward speed is 1100 m/h, the power consumption increases with the increase of tool roller speed and bending angle.

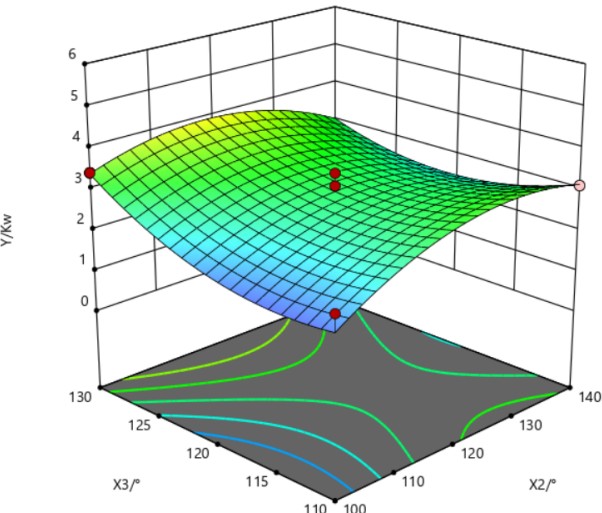

**Figure 11.** $X_2X_3$ interaction response surface.

### 5.3. Field Test Validation

The experiment was conducted in Wugong Village, Shihezi City, Xinjiang in May 2022. The average soil firmness was 2.16 MPa and the moisture content was 10.64%. The supporting power was a TN654 tractor, and the test equipment included: a rotary tillage device, a mechanical tachometer (measuring range 0~400 r·min$^{-1}$), and NJTY3 general dynamic telemetry system of agricultural machinery. The power consumption measurement system and field rotary tillage operation are shown in Figure 12.

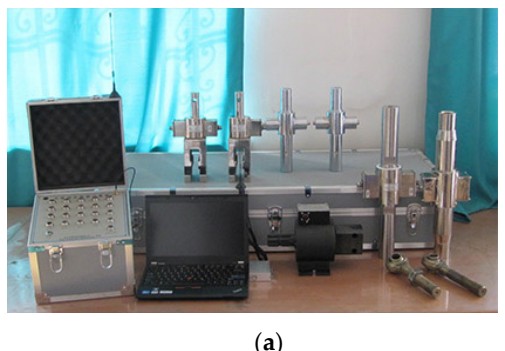
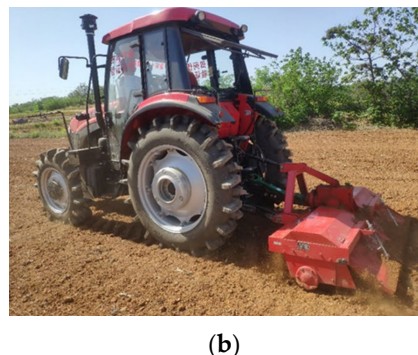

(**a**)                                                                    (**b**)

**Figure 12.** Field test. (**a**) general dynamic telemetry system for agricultural machinery; (**b**) field operation device.

The power consumption measurement used wireless telemetry technology [26,27] and the technical scheme of the integrated torque sensor of the power output shaft and frameless three-point suspension traction sensor. The measurements from the initial simulation parameters showed a forward speed $v$ = 1100 m/h, speed $n$ = 120 r/min, and actual power consumption of 1.71 kW. At the same time, field tests consistent with the parameters of the above 17 simulation tests were conducted, and data J was obtained for comparison, as

shown in Figure 13. The actual power consumption was higher than the simulation value because the actual operation was caused by the extra power consumption caused by the root system, debris, and other friction and wear in the soil. However, the simulation test error was less than 12%, which verifies the accuracy of the simulation test and has reference value to establish the power model.

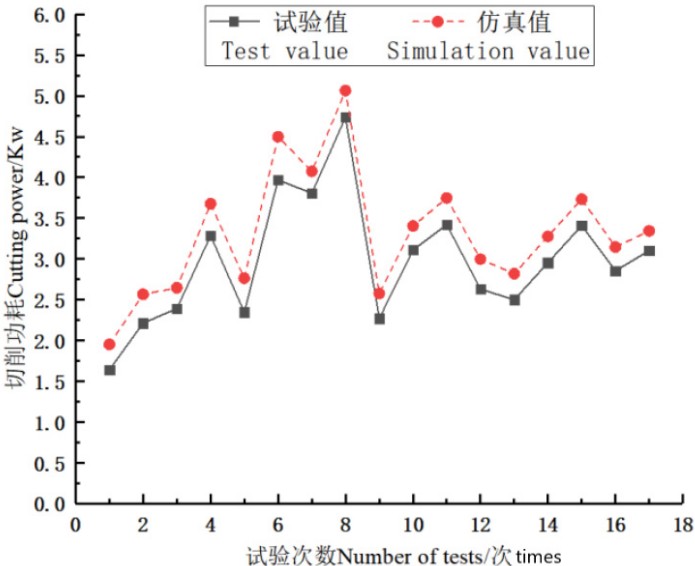

**Figure 13.** Comparison of power consumption in field trials and simulation trials.

The soil disturbance during cutting and the soil appearance after cutting were similar to the results of discrete element simulation, which further verified the accuracy of discrete element simulation.

*5.4. Experimental Optimization Results*

When the forward speed was 800~1400 m/h, the cutter roll speed was 100~140 r/min, and the bending angle was 110~130°. In order to obtain the optimal parameter solution, the optimal protection design was carried out. The objective the boundary condition parameter model of optimization and experimental factors was established as follows:

$$\min Y \begin{cases} 800 \leq X_1 \leq 1400 \\ 100 \leq X_2 \leq 140 \\ 110 \leq X_3 \leq 130 \\ 1.24 \leq Y\,(X_1,\,X_2,\,X_3) \leq 4.763 \end{cases} \tag{7}$$

In the prospect of low energy consumption and high efficiency in agricultural machinery operation [28,29], considering the influence of finite-element simulation structure, performance, and practical application production, the optimal factor parameter combination was finally determined as follows: forward speed 900 m/h, rotation speed 100 rad/min, bending angle 115°, and power consumption 1.458 kW. The simulation experiment was carried out when the optimal combination parameters were set. The calculated working resistance was 1.64 kW, which was basically consistent with the optimization results. Meanwhile, the finite-element evaluation was carried out and the calculated results met the mechanical requirements of rotary blade.

Finally, a field test was carried out to verify the working resistance of 1.83 kW. Compared with the rotary tillage power under the initial parameter setting, the power consumption was reduced by 10%, which proves that the optimization parameters reduce the actual cutting power consumption of the cutter roll, which has reference value for power optimization.

## 6. Conclusions

(1) The bending angle is one of the key structural parameters that affect the performance of the rotary blade. When the bending angle is 110–130°, the structural stress and bending deformation increase with the increase of the bending angle. When the bending angle is 110°, 120°, and 130°, the average deformation is 0.94 mm, 1.13 mm, and 1.22 mm, and the average stress is 39.67 Mpa, 45.39 Mpa and 50.98 Mpa, respectively. The analysis results provide reference for the subsequent structural parameter optimization.

(2) Through the use of the discrete-element simulation method, we carried out cutting simulation and process analysis. Using the design of experiments method to establish the tool's roll cutting power consumption, the fitting equation and mathematical model can better predict the power consumption. The established cutting power fitting equation and mathematical model can better predict power consumption. Considering the structural performance, actual production, and application of the rotary blade, the optimal parameters were determined as follows: forward speed 899.96 m/h, rotation speed 100.93 r/min, and bending angle 116.43°. The corresponding minimum power consumption was 1.44 kW, the average deformation of a single rotary blade was 0.983 mm, and the average stress was 41.826 Mpa, which were reduced by 15.8%, 13%, and 7.9%, respectively. Through the comparison of the field experiment and 17 groups of simulation experiment data, the error of the actual power consumption and the simulation experiment was lower than 12%, and the structural performance met the requirements of rotary tillage intensity.

(3) The optimal parameters obtained from experiment optimization were verified by the simulation experiment and the field experiment. The power consumption was 1.55 kW and 1.6 kW respectively, which was 9.4% and 6.4% lower than that under the initial parameters. The structural performance of the optimized rotary blade was evaluated by the finite-element method, and the calculated results met the mechanical requirements of the rotary blade. The overall optimization results achieved effective reduction of power consumption, in line with the green development direction of agricultural machinery.

**Author Contributions:** Conceptualization, X.Z. and L.Z.; methodology, X.Z.; software, X.Z.; validation, X.S., X.M. and H.W.; formal analysis, H.W.; investigation, X.Z.; resources, X.Z.; data curation, X.Z., X.H. and H.W.; writing—original draft preparation, X.Z.; writing—review and editing, L.Z. All authors have read and agreed to the published version of the manuscript.

**Funding:** This research was funded by the Construction Project of Demonstration Platform for National New Materials Production and the Application-Demonstration Platform for Production and Application of Materials on Agricultural Machinery Equipments, grant number TC200H01X-5.

**Institutional Review Board Statement:** Not applicable.

**Informed Consent Statement:** Not applicable.

**Data Availability Statement:** All relevant data presented in the article are stored according to institutional requirements and, as such, are not available on-line. However, all data used in this manuscript can be made available upon request to the authors.

**Conflicts of Interest:** The authors declare no conflict of interest.

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
