# Peer review of "Simulation of Soil Cutting and Power Consumption Optimization of a Typical Rotary Tillage Soil Blade"

_applsci, doi:10.3390/app12168177_

Round 1

Reviewer 1 Report

Thank you for the interesting paper, I have a few remarks:

1. Text. Can you check the text to keep the spaces between the words. Check for duplicate characters, such as “3” on page 5 (line 136).

2. Figure 6b. If you use a linear elastic-plastic model, mention it. Check if the arrow should point in the opposite direction during loading? Select the text in the image: "Load, Unload/Reload" or "Loading, Unloading/Reloading"

3. Equation 4. Check the text describing equation.

Author Response

请参阅附件

Reviewer 2 Report

The author presented work on the FEM and DEM simulation of rotary cutting parts. The idea of research is interesting however there are some suggestions to modify the manuscript. please see the attached file
